*Method*

# An integrated workflow for crosslinking mass spectrometry

Marta L Mendes[1,†,‡] (iD), Lutz Fischer[1,2,‡] (iD), Zhuo A Chen[1], Marta Barbon[3,4] (iD), Francis J O'Reilly[1] (iD),
Sven H Giese[1], Michael Bohlke-Schneider[1], Adam Belsom[1,2], Therese Dau[2], Colin W Combe[2],
Martin Graham[2], Markus R Eisele[5], Wolfgang Baumeister[5], Christian Speck[3,4] & Juri Rappsilber[1,2,*] (iD)

## Abstract

We present a concise workflow to enhance the mass spectrometric detection of crosslinked peptides by introducing sequential digestion and the crosslink identification software xiSEARCH. Sequential digestion enhances peptide detection by selective shortening of long tryptic peptides. We demonstrate our simple 12-fraction protocol for crosslinked multi-protein complexes and cell lysates, quantitative analysis, and high-density crosslinking, without requiring specific crosslinker features. This overall approach reveals dynamic protein–protein interaction sites, which are accessible, have fundamental functional relevance and are therefore ideally suited for the development of small molecule inhibitors.

**Keywords** crosslinking mass spectrometry; protein–protein interactions; proteomics; software; structural biology

**Subject Categories** Methods & Resources; Proteomics

**Mol Syst Biol. (2019) 15: e8994**

## Introduction

Crosslinking mass spectrometry (CLMS) has become a standard tool for the topological analysis of multi-protein complexes and has begun delivering high-density information on protein structures, insights into structural changes and the wiring of interaction networks *in situ* (O'Reilly & Rappsilber, 2018). The technological development currently focuses on enrichment strategies for crosslinked peptides and mass spectrometric data acquisition (Leitner *et al*, 2013; Kolbowski *et al*, 2017; Liu *et al*, 2017), including newly designed crosslinkers (Kao *et al*, 2011). MS2-cleavable crosslinkers, in particular, have celebrated recent

successes for the analysis of protein complexes (Wang *et al*, 2017) or complex mixtures (Chavez *et al*, 2013; Liu & Heck, 2015).

The focus on bespoke crosslinkers has left general steps of sample preparation, such as protein digestion, with less attention. Tryptic digestion generates crosslinked peptides of considerable size, a quality that has been exploited with their enrichment by SEC (Leitner *et al*, 2012b), but one that poses as a potential problem regarding their detection. Replacing trypsin with proteases such as GluC, AspN and chymotrypsin does not change peptide size distributions fundamentally (Swaney *et al*, 2010). We reasoned that sequential digestion could reduce the size of large tryptic peptides and offer access to sequence space that otherwise would remain undetected. We therefore followed trypsin digestion with subsequent digestion by alternative proteases and developed xiSEARCH, a database search engine, allowing the search of multiple datasets resulting from the application of our protocol. This novel approach expands the detectable structure space in proteins, allowing it to capture dynamic regions in protein complexes that are mechanistically important and therefore *a priori* druggable, however that hitherto have remained undisclosed by cryo-EM due to their flexible nature.

## Results

### Sequential digestion increases the number of identified crosslinks

We first tested this workflow on a standard mix of seven Bis[sulfosuccinimidyl] suberate (BS[3]) crosslinked proteins (catalase, myoglobin, cytochrome C, lysozyme, creatine kinase, HSA and conalbumin). Importantly, their structures are known and hence offer an independent assessment of false identifications. Four digestion conditions, each giving three SEC fractions, resulted in a total

1   Bioanalytics, Institute of Biotechnology, Technische Universität Berlin, Berlin, Germany
2   Wellcome Centre for Cell Biology, University of Edinburgh, Edinburgh, UK
3   MRC London Institute of Medical Sciences (LMS), London, UK
4   DNA Replication Group, Faculty of Medicine, Institute of Clinical Sciences (ICS), Imperial College London, London, UK
5   Department of Molecular Structural Biology, Max Planck Institute of Biochemistry, Martinsried, Germany
    *Corresponding author. Tel: +49 30 314-72374; E-mail: juri.rappsilber@tu-berlin.de
    ‡These authors contributed equally to this work
    †Present address: Quantitative Biology Unit, Luxembourg Institute of Health, Luxembourg, Luxembourg

of 12 acquisitions, which is the protocol applied to all subsequent analyses presented here (Fig 1A). The results of this protocol for our standard proteins were compared to a parallel digestion using the same four enzymes and using trypsin alone in four replica, maintaining the analytical effort comparable in all three cases (SEC fractionation, 12 injections). Sequential digestion produced the best results when compared to replica analyses and parallel digestion (Figs 1B and C, and EV1, Dataset EV1). Before assessing if this improvement translated into a gain of information in biological applications, we investigated the origin of the added data (Figs EV2 and EV3, Dataset EV4).

Indeed, sequential digestion led to smaller peptides than trypsin alone (Figs 1D and EV2F, and Dataset EV4) and moved the mass distribution of theoretical crosslinkable peptides more into the mass range typically detected by our instrument (Fig EV2F, Dataset EV4). For short peptides, we noticed a protection effect, based on the number of peptides containing missed cleavage sites and on the number of missed cleavage sites relative to peptide length (Figs EV3 and EV2B, and Dataset EV4). This agrees with reports that serine proteases lose efficiency as peptides shorten (Thompson & Blout, 1973; Wenzel & Tschesche, 1981). Although AspN is a metalloprotease, it showed a similar loss of efficiency for short peptides. Notably, we observed a bias towards maintaining tryptic C-termini. Crosslinked peptides with two tryptic C-termini are more frequently

identified while those with C-termini generated by the second protease are less frequent than expected, relying on N-termini as internal reference (Fig EV4). This identification bias is consistent with better fragmentation behaviour of peptides with basic C-termini (Olsen *et al*, 2004) and testifies to the importance of trypsin as part of the protocol.

We then tested the sequential digestion approach on samples of increasing complexity ranging from single proteins, UGGT and C3b, to the OCCM DNA replication complex (1.1 MDa), the 26S proteasome (2.5 MDa) and high-molecular weight fractions of human cytosol. A quantitative experiment was performed to assess the efficiency of sequential digestion combined with the QCLMS workflow (Chen *et al*, 2016c). Additionally, we tested the approach using two different crosslinkers, the homobifunctional crosslinker BS³ and the heterobifunctional, photoactivatable crosslinker sulfosuccinimidyl 4,4′-azipentanoate (SDA).

## Compatibility with photo- and quantitative CLMS

UGGT was one of the data-assisted *de novo* folding targets of CASP12 for which we contributed data in the form of 433 unique residue pairs obtained at a 5% FDR (http://predictioncenter.org/download_area/CASP12/extra_experiments/; Appendix Fig S1A) using SDA as crosslinker and 26 LC-MS runs (Ogorzalek *et al*,

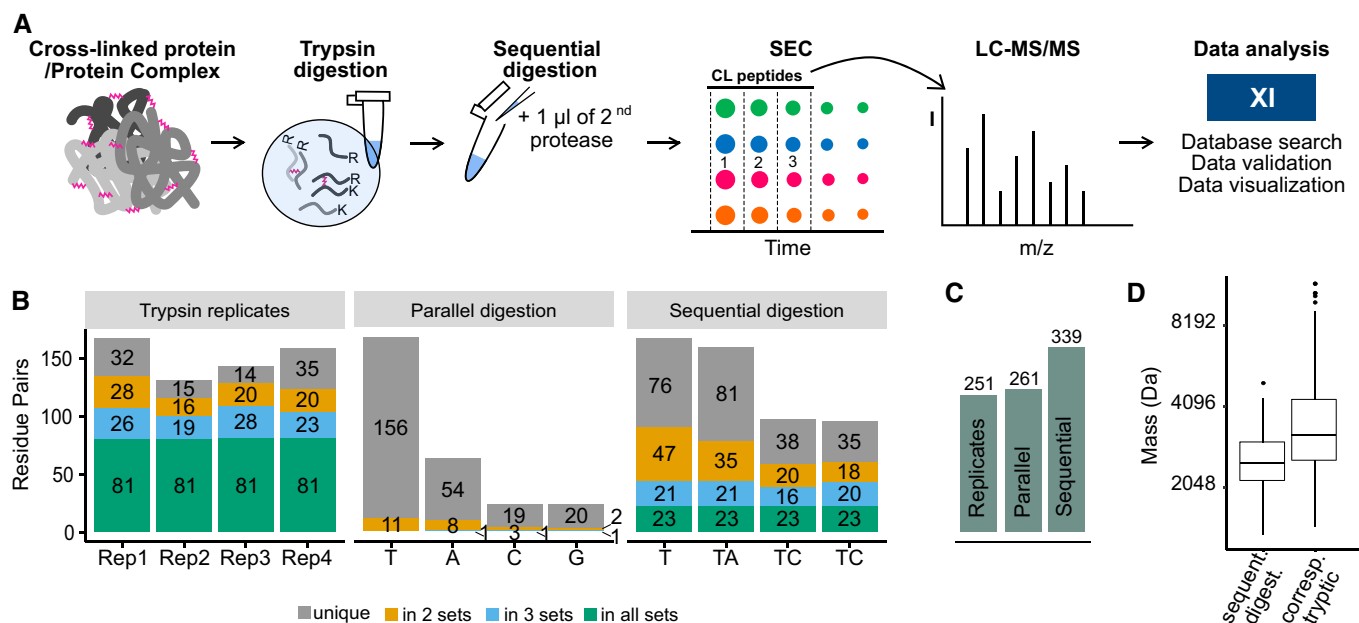

**Figure 1. Sequential digestion workflow compared to repeated analysis and parallel digestion.**

A   Sequential digestion workflow. Proteins or protein complexes are crosslinked and digested with trypsin. After splitting the sample into four aliquots, one remains single digested with trypsin (T) while the others are sequentially digested with either AspN (A), chymotrypsin (C) or GluC (G). Samples are enriched by SEC, and the three high-MW fractions are analysed by LC-MS, submitted to xiSEARCH and xiFDR analysis.

B   Results of the sequential digestion workflow applied to a synthetic 7-protein mix, compared to using trypsin alone in four replicates and parallel digestion with trypsin, AspN, chymotrypsin and GluC. A trypsin four replicate experiment shows a large overlap of the four datasets with little gain. Parallel digestions with trypsin, AspN, chymotrypsin and GluC demonstrate high complementarity but moderate gains over trypsin. Sequential digestion shows low overlap between the four datasets and the largest gain in unique residue pairs.

C   Gains of repeated analysis (trypsin only), parallel digestion and sequential digestion for the same data as shown in panel (B).

D   Crosslinked peptides obtained by sequential digestion of a synthetic 7-protein mix are smaller than their corresponding tryptic peptides. Boxplot ranges represent the 25th (lower hinge) and 75th (upper hinge) percentiles, respectively. Middle line represents the median. For trypsin 4 replicates were analysed and for sequential digestion and parallel digestion 1 sample was analysed.

2018). Using sequential digestion, we now identified 1,523 unique residue pairs in only 12 runs (Appendix Fig S1B and C, Dataset EV1 and EV2). With 5% long-distance links (> 20 Å) when mapped onto the structure released by CASP organisers (Appendix Fig S1D), the 300% increase in observed links comes at uncompromised reliability. Consequently, the sequential digestion protocol improves high-density CLMS by a clear increase in the number of residue pairs while simultaneously reducing the analytical effort needed to detect these.

We next combined quantitative CLMS (QCLMS; Schmidt *et al*, 2013; Tomko *et al*, 2015; Chen *et al*, 2016b) with our workflow (Appendix Fig S2A) to investigate the dimerisation of C3b. Thioester-mediated dimerisation of C3b is a key process of the human complement response enhancing the efficiency of C5 convertase formation which ultimately leads to clearance of pathogens from human blood (Hong *et al*, 1991; Rawal & Pangburn, 2001; Pangburn & Rawal, 2002). However, the structure of this dimer is currently unclear. The reactive thioester could result in a random orientation of the two C3b molecules in a dimer. Alternatively, auxiliary factors or self-organisation properties of C3b could mediate a preferred orientation. We here investigate C3b alone and find it to form dimers in the absence of active thioester and auxiliary proteins. We quantified 293 unique crosslinks, about three times more than with trypsin alone (99) (Appendix Fig S2B–D; Dataset EV1 and EV2) which lends robust support to a bottom-to-bottom orientation (Appendix Fig S2E). This suggests non-covalent interactions between C3b molecules lead to a preferred dimer orientation which implies that a thioester bridged dimer would follow this arrangement. Non-covalent interactions thus self-organise C3b into a productive dimer as this arrangement is compatible with the subsequent molecular events of the complement cascade by allowing unhindered binding of factor B at the top of C3b.

### A novel and functionally important contact in the OCCM complex

Turning our attention to protein complexes, we investigated the OCCM complex, a helicase loading intermediate formed during the initiation of DNA replication. Recently, a 3.9-Å structure of *Saccharomyces cerevisiae* OCCM on DNA was obtained by cryo-electron microscopy (cryo-EM), supported by CLMS (Yuan *et al*, 2017). We identified 682 residues pairs from the same sample analysed before, with large contribution from sequential digestion (Fig 2A and Appendix Fig S3A–C; Dataset EV1 and EV2). Interactions observed now include known Cdt1-Mcm2 and Mcm6 but also Mcm2-Orc5 interaction (Mcm2-850-Orc5-369). These led us to delete the C-terminal 20 aa of Mcm2 (848–868; Fig 2B, lane 5) and analyse its biological relevance in a well-established *in vitro* helicase loading assay, which recapitulates the *in vivo* process (Evrin *et al*, 2009). The deletion mutant did not affect ORC, Cdc6, Cdt1 and origin DNA-dependent complex assembly under low salt conditions (Fig 2B, lanes 6 and 7), but severely impaired the formation of the high salt stable double-hexamer (Fig 2B, lanes 8 and 9), the final product of the helicase loading reaction. This is an exciting result, as it highlights a novel and essential role for Mcm2 C-terminus in a late step of MCM2-7 double-hexamer formation (Barbon *et al*, in preparation), a process that is only poorly understood. Moreover, the CLMS data show that the Mcm2 C-

terminus is involved in a network of interactions with flexible domains of Orc6, Orc2 and Mcm5, indicating dynamics at the Mcm2-Mcm5 DNA entry gate (Samel *et al*, 2014), which could represent an ideal target for the development of inhibitors with potential as anti-cancer therapy (Gardner *et al*, 2017), as dynamic interactions have improved druggability characteristics over stable protein interactions (Ulucan *et al*, 2012; Jubb *et al*, 2015). Indeed, expressing Mcm2-7ΔC2 causes dominant lethality (Fig 2C). The ability of CLMS data to complete the cryo-EM structure of the OCCM complex by providing information on dynamic contacts proved here essential. Note that 15% of our residue pairs falling into the published OCCM structure were long distance (> 30 Å, Appendix Fig S3D). This indicates that CLMS unveils dynamic aspects of protein complex topology also in regions of the structure accessible to cryo-EM as will become even more evident in our proteasome analysis.

### Conformational diversity of the 26S proteasome

We next analysed an affinity-purified 26S proteasome sample, containing more than 600 proteins (Dataset EV3). The results of our workflow compare favourably with the largest analysis reported on this complex to date (Wang *et al*, 2017) in terms of numbers ($n = 1,644$ vs. 447 unique residue pairs in the proteasome at 5% FDR; Fig 3A and B, Appendix Fig S4A and B, Dataset EV1 and EV2) and in terms of agreeing with the structure of the individual subunits (6% vs. 26% long-distance links (> 30 Å); Fig 3A and B). Links between proteins ($n = 602$) reveal a large amount of topological variability in the proteasome, with 30% ($n = 179$) being not covered by current cryo-EM-based models and thus extending our awareness of the proteasome structure to more dynamic regions. Long-distance links ($n = 191$ between and 85 within proteins) are mainly distributed in the base of the proteasome, where ATP binding and hydrolysis lead to a large conformational variety (Fig 3C, Appendix Fig S4C and D). Indeed, some of these links ($n = 78$) not matching to one structure of the proteasome mapped well to alternative conformational states stabilised by ATP analogue (Unverdorben *et al*, 2014; Wehmer *et al*, 2017). State-specific crosslinks were found predominant in the AAA-ATPase-dependent heterohexameric ring (Fig 3D–G Appendix Fig S4C and D) indicating rearrangement of Rpn5 relative to Rpt4 (Fig 3D). In the s2 state, our data support Rpn1 being translated and rotated to be positioned closer to the AAA-ATPase (Fig 3E). Crosslinks therefore support in solution the cryo-EM-based model of substrate transfer to the mouth of the AAA-ATPase heterohexameric ring (Unverdorben *et al*, 2014) and point towards the existence of additional conformational states that remain to be defined to fully understand the complex's function and that may offer as conformer-specific interactions prime intervention points for small molecule inhibitors.

### Exploration of the human cytoplasm

To probe our 12-fraction protocol in large-scale CLMS, we analysed seven high-molecular weight fractions of human cytosol. We identified 3,572 unique residue pairs (5% FDR, 528 proteins, Figs 4A, and EV5A and B, Dataset EV1 and EV2). This is in line with recent studies reporting 1,663 and 3,045 unique residue pairs, respectively,

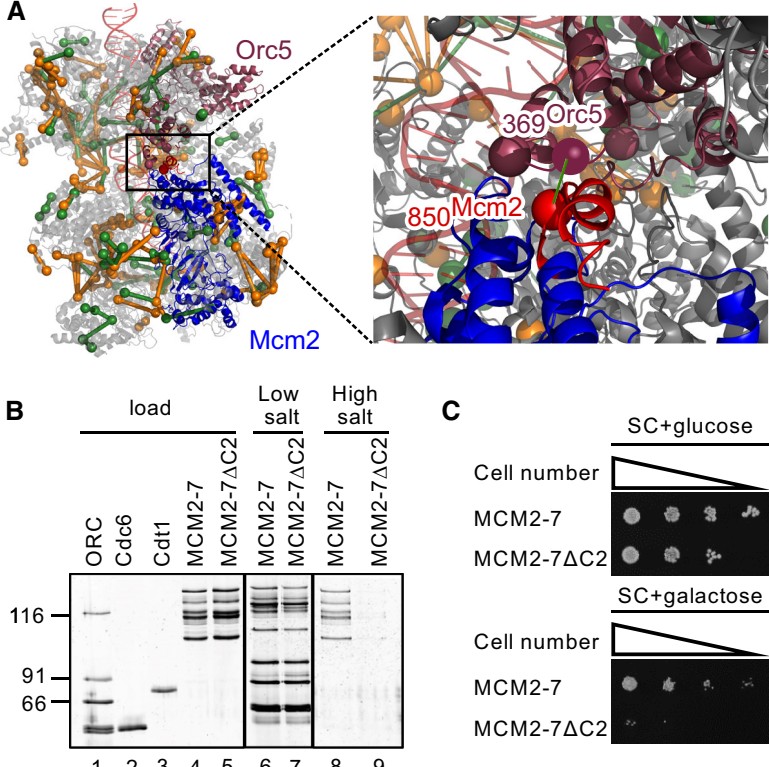

**Figure 2. Sequential digestion of the affinity-purified complexes OCCM (*Saccharomyces cerevisiae*; Residue pairs observed in tryptic (green) and non-tryptic (orange) peptides).**

A Unique residue pairs mapped to the OCCM complex (PDB 5UDB) and the key link Mcm2-Orc5 (Mcm2-850-Orc5-369).

B The *in vitro* helicase loading assay demonstrates that an Mcm2 C-terminal deletion mutant supports complex assembly (lanes 6 and 7) and blocks formation of the final helicase loading product (lanes 8 and 9).

C Overexpression analysis of Mcm2-7ΔC2 shows that this mutant causes dominant lethality, indicating that the C-terminus of Mcm2 is essential in cell survival.

albeit using a cleavable crosslinker and analysing whole cell extracts (Liu *et al*, 2015, 2017). While the overlap between the published data and ours is low (Fig 4A), this will be influenced by factors such as the different starting material and the different analytical strategies.

Our protein–protein interaction network included previously observed complexes such as the Mcm2-7 complex, the 26S proteasome, the ribosome, the COPI complex, the TRiC-CCT complex and the HS90B-CDC37-Cdk4 complex (Figs 4B and EV5C–G, and Appendix Fig S5). For the 26S proteasome, we were able to distinguish between different states defining flexibility in the AAA-ATPase ring (Fig 4C; Chen *et al*, 2016a). This indicates the ability of our protocol to unveil dynamic interactions in mixtures nearing the native environment complexity of proteins.

## xiSEARCH, identification of crosslinks from mass spectra

To analyse the mass spectrometric data of these and other studies (Liu *et al*, 2015, 2017; Wang *et al*, 2017), we developed our database search software xiSEARCH (Figs 5A and EV6, Appendix Tables S1 and S2, https://rappsilberlab.org/software/). The algorithm of xiSEARCH has been described conceptually before (Giese *et al*, 2016). It follows an approach that computationally unlinks crosslinked peptides and by doing so circumvents the n2 database problem of crosslinking. Like pLink 2 (Chen *et al*, 2019), StavroX (Götze *et al*, 2012) and Kojak (Hoopmann *et al*, 2015), xiSEARCH allows to search any crosslink and protease specificity; thus, xiSEARCH's performance was assessed against these three alternatives (Fig 5B). StavroX became non-responsive and was not pursued further. Kojak was paired with PeptideProphet (Keller *et al*, 2002) and xiSEARCH with xiFDR (Fischer & Rappsilber, 2017) to maximise results and control the error rate. We assess the error on the level of unique residue pairs—as this is the actual information of interest. This is native to XiFDR while for Kojak(+PeptideProphet) and pLink 2 the output was sorted by score on the level of PSMs, only the best scoring PSM per residue pair was kept and a 5% FDR on the, now unique residue pairs calculated. By default, xiSEARCH weights the likelihood of a K vs. S, T or Y being involved in a crosslink higher to reduce the number of unique links without strong support by data. For the purpose of comparison, xiSEARCH was run with this feature enabled (marked xiSEARCH*) and with this feature disabled, as none of the other tools are supporting a similar consideration. xiSEARCH reports 91% more unique links than Kojak + PeptideProphet and 45% more than pLink 2 (Fig 5B, Appendix Fig S6). Note that we and

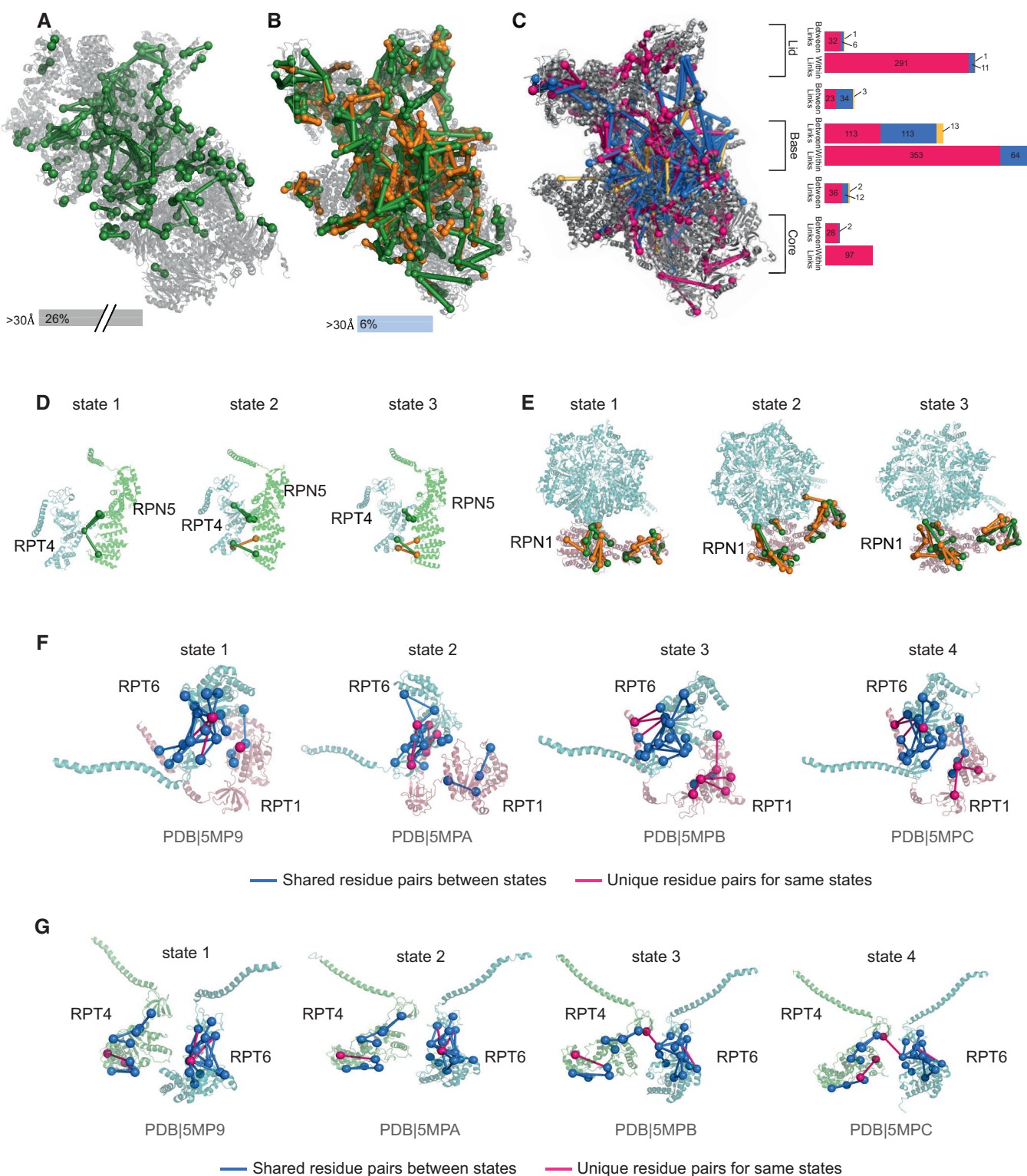

**Figure 3.**

the developers of these tools could not pair Kojak with Peptide-Prophet for sequential digestion. To check the reliability of residue pairs uniquely reported by xiSEARCH, we assessed them for the structurally rigid proteasome core particle. Less than 3% were found to be long distance and thus very plausibly false, which is in good agreement with the expected FDR of 5%. In summary, xiSEARCH performed very favourably compared to other universal software for CLMS.

**Figure 3. Sequential digestion of the 26S proteasome from *Saccharomyces cerevisiae*.**

A   Unique residue pairs obtained by Wang *et al* for the human 26S proteasome (PDB 5GJR).

B   Unique residue pairs obtained by sequential digestion for the *S. cerevisiae* 26S proteasome (PDB 4CR2). Sequential digestion returned the highest number of residue pairs so far identified by CLMS for the 26S proteasome. Tryptic residue pairs are represented in green and non-tryptic in orange.

C   Long distance (blue) and within distance (pink) between residue pairs were mapped into one of the states of the proteasome (4CR2) showing the accumulation of those into the base of the complex. Residue pairs satisfying other states are represented in yellow. The bar plot shows the distribution of all residue pairs in the complex showing that long-distance links locate mainly in the base.

D   Unique residue pairs were mapped into the three states described by Unverdorben *et al* showing the rearrangement of Rpn5 relative to Rpt4.

E   Our data support Rpn1 being translated and rotated to be positioned closer to the AAA-ATPase.

F   Structural rearrangements of the AAA-ATPase-dependent heterohexameric ring throughout four states for the RPT6 and RPT1 mapped to the four states described by Wehmer *et al.*

G   Structural rearrangements of the AAA-ATPase-dependent heterohexameric ring throughout four states for the RPT4 and RPT6.

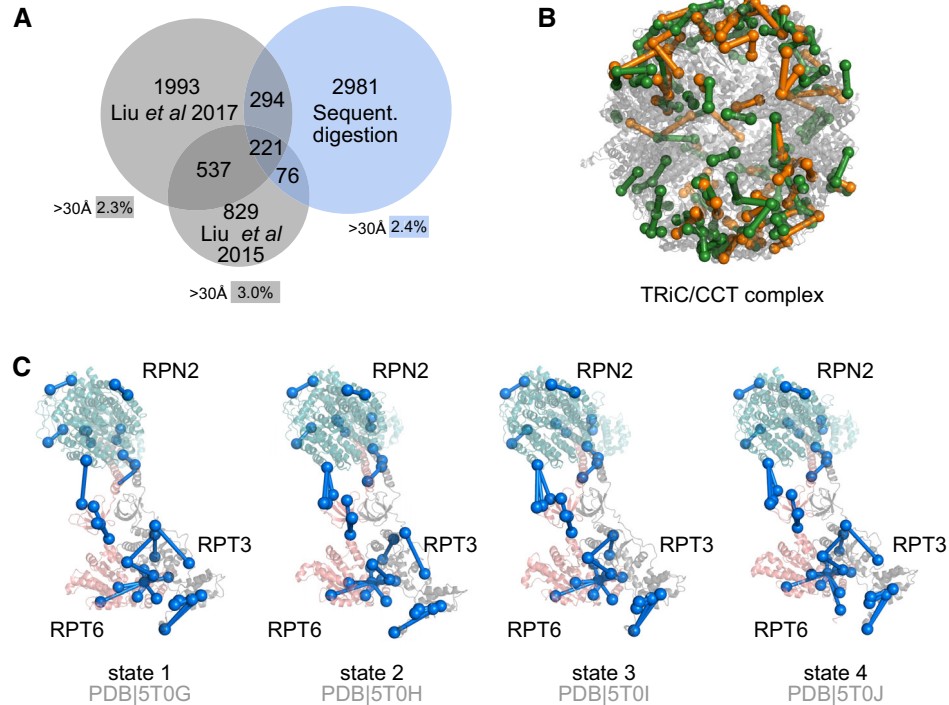

**Figure 4. Sequential digestion of the 26S proteasome from human cytosol.**

A   Comparison of sequential digestion for complex mixtures with previous studies. Sequential digestion returns a higher number of residue pairs with low overlap to published datasets, showing the complementarity of the different approaches. Long-distance links were determined only within proteins and are comparable for all datasets.

B   Residue pairs for the TRiC/CCT complex were mapped into the crystal structure and support the rearrangement of the complex reported by Leitner *et al* (2012a) (PDB 4V94).

C   Despite the complexity of the sample, we were able to identify the four states of the 26S proteasome showing the flexibility of the AAA-ATPase-dependent heterohexameric ring.

## Discussion

### Sequential digestion novelty

The use of proteases other than trypsin to achieve complementary information in protein analysis dates back to at least 1987 (Aebersold *et al*, 1987). Since then, several works used parallel digestion and fewer used sequential digestion to increase sequence coverage and/or complementarity in simple protein complexes (Mohammed *et al*, 2008), proteomes (MacCoss *et al*, 2002; Swaney *et al*, 2010;

Guo *et al*, 2014), phosphoproteomes (Wang *et al*, 2008; Gilmore *et al*, 2012; Giansanti *et al*, 2015) and other post-translational modifications (Larsen *et al*, 2005). In CLMS, parallel digestion was reported first by Pinkse *et al* (2009) to increase complementarity and target a specific crosslink site. Leitner *et al* in 2012 presented crosslink data obtained when using in parallel five different proteases on a mix of standard proteins. Unfortunately, it remained unclear if the parallel use of five proteases would have been outperformed by simply five times re-analysing a tryptic digest. Re-analysing tryptic digests results in additional crosslinks being

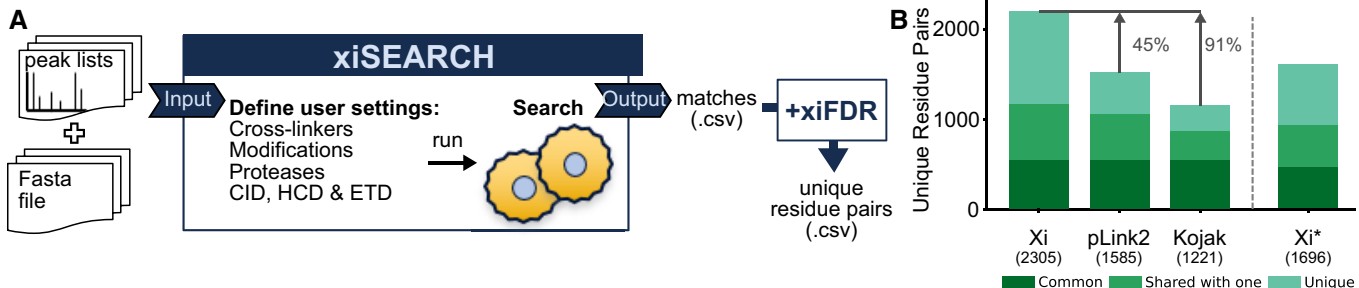

**Figure 5. xiSEARCH.**

A   xiSEARCH is an open source search engine that takes a peak list as input. Users can define any type of crosslinker, modification, digestion and fragmentation method. The output is a list of matches in .csv format. We use xiFDR to filter results to the desired confidence level.

B   xiSEARCH + xiFDR(Xi), pLink 2 and Kojak(+PeptideProphet) comparison at 5% residue-pair FDR. The same trypsin dataset of the 26S proteasome was searched with all three software packages. xiSEARCH was run twice—once giving same likelihood for matching lysine, serine, threonine and tyrosine (xiSEARCH), as is the case for Kojak and pLink 2, and once giving priority to Lysine (Xi*).

detected (Müller *et al*, 2018) also seen here (analysing a trypsin digest four times increased the observation from 167 to 251 unique residue pairs (URPs), Fig 1B and C). We then analytically prove that parallel digestion outperforms repeated injection of tryptic digests for crosslink detection, albeit moderately (261 vs. 251 URPs, Fig 1C). Importantly, we expand this by showing that sequential digestion is even better (339 URPs, Fig 1C).

We underpin these observations by proposing a mechanistic model that explains why sequential digestion outperforms parallel and repeated use of proteases. Outperforming repeated use of trypsin is linked to sequentially digested crosslinked peptides being smaller (Fig EV2A, Dataset EV4) which in fact improves detection rates as most observed peptides fall between 1,000 and 2,000 Da (Fig EV3D). The second digestion targets primarily long peptides as can be seen both in the surviving long peptides (poor in secondary cleavage sites) as well as in the observed short peptides (rich in secondary cleavage sites; Fig EV3). Consequently, sequential digestion is effective and does not shorten crosslinked peptides such that they become less observable (below 1,000 Da). Outperforming parallel digestion is linked to the detection bias of proteomics towards peptides with tryptic C-termini, that has been noted for linear peptides (Giansanti *et al*, 2016). Cleaving a crosslinked peptide N-terminal of the crosslink site maintains the tryptic C-termini while cleaving C-terminal leads to a non-tryptic C-terminus. During digestion, both should be equally likely. However, the former will be more likely to be detected which is reflected in our data (Fig EV1C). As all crosslinked peptides of parallel digestion other than those of trypsin use are lacking tryptic C-termini, parallel digestion has a marked disadvantage.

### Search software comparison

In our hands, xiSEARCH performs favourably when compared with other crosslink search software packages. However, a software comparison involving only the developers of one of the software packages will likely always be incomplete. A balanced evaluation of the merits of crosslink search tools would ideally be based on a community-wide effort with contributions of the developers of the respective software packages.

Here, we were restricted to software tools that supported sequential digestion and that supported a comparable set of features as used in our data analysis. One such point is that we search BS3 with specificity to lysine, serine, threonine and tyrosine. This already excludes many crosslink software packages, due to their restriction to lysine only. Furthermore, we place linkage sites in the absence of strong distinctive evidence preferentially on lysine. To our knowledge, this is not supported by any other crosslink search software. For the comparison, we removed that preference—likely inflating false results (Appendix Fig S6).

### Protocol applicability

Most published protocols are tested only for specific samples and application areas. Furthermore, workflows tend to build on one specific combination of crosslinker and database search tool. In contrast, we deliver a protocol with broad applicability, demonstrating its use in protein dynamics, protein complexes topology, conformational changes (qCLMS) and using homo- and heterobifunctional crosslinkers. Our integrated workflow utilises standard crosslinkers, without special chemistries to assist analysis. This permits the sequential digestion workflow to be combined with other crosslinkers such as MS-cleavable crosslinkers. Recent large-scale studies have successfully used MS-cleavable crosslinkers (Liu *et al*, 2015; Hage *et al*, 2017). Note, however, that our work uses standard crosslinkers at no obvious disadvantage. Importantly, MS-cleavable crosslinkers have yet to be combined with high-density crosslinking and are likely incompatible with crosslinking by non-canonical amino acids (Suchanek *et al*, 2005) motivating efforts in keeping crosslink chemistry and analysis workflows separate. Our protocol supports this drive and provides a concise, universal protocol to increase data density and ease of use for CLMS in diverse applications. In particular, we were able to identify dynamic protein interaction regions and topologies which are notoriously difficult to detect using conventional structural biology methods due to their flexibility yet are prime therapeutic intervention points, as these important interactions are only short-lived and therefore druggable.

## Complementarity of CLMS to cryo-electron microscopy

Crystallography and cryo-EM employ averaging approaches to obtain high-resolution structural information of proteins or protein complexes. Structural heterogeneity is not compatible with crystallography, while novel cryo-EM data analysis approaches can deal with some conformational heterogeneity. Nevertheless, cryo-EM also fails to visualise flexible regions.

Indeed, in the context of the OCCM, cryo-EM revealed the overall organisation of the 14 subunit complex. However, in an area of high protein flexibility, which encompasses the important Mcm2/Mcm5 DNA entry gate of the MCM2-7 ring, the EM structure was hampered by low resolution. By CLMS, a new network of interactions was detected at the DNA entry gate, involving Mcm2, Mcm5, Orc2 and Orc6. Moreover, we showed that the Mcm2 interaction with Orc6 is functionally relevant, as a small Mcm2 deletion in the interaction surface strongly affected helicase loading and caused lethality *in vivo*. As such, CLMS and cryo-EM can synergise to identify and characterise flexible regions in protein complexes, which have important functionality.

Interestingly, protein–protein interactions identified by cryo-EM and crystallography are nearly impossible to target by small molecules, since they are very long-lived and due to their hydrophobicity not accessible for water-soluble small molecules. On the other hand, CLMS, employing water-soluble crosslinkers, naturally identifies well-hydrated dynamic protein–protein interaction surfaces, which are typically less stable or context-dependent and therefore ideally suited for drug development. In summary, CLMS is uniquely capable of detecting flexible and dynamic protein interactions, which makes the technology highly synergistic with other structural approaches and opens a window of opportunity for drug development of dynamic protein–protein interactions.

# Materials and Methods

**Reagents and Tools table**

| Reagent/resource | Reference or source | Identifier or catalogue number |
|---|---|---|
| **Experimental models** | | |
| YYS40 (*S. cerevisiae*) | Sakata, E. *et al* | Prof. Dr. Wolfgang Baumeister, MPI, Martinsried |
| K562 cells (*Homo sapiens*) | DSMZ | #ACC-10 |
| OCCM complex | Evrin, C. *et al* | Prof. Dr. Chistian Speck, MRC-LMS or ICS-ICL, London |
| **Proteins** | | |
| Catalase | SIGMA | C9322 |
| Myoglobin | SIGMA | M1882 |
| Cytochrome C | SIGMA | C2037 |
| Lysozyme | SIGMA | L6876 |
| Human serum albumin | SIGMA | A8763 |
| Conalbumin | SIGMA | C0755 |
| Creatine kinase | Roche | 10127566001 |
| C3b | Complement Technology, Inc. | A114 |
| UGGT | CASP 12 | Prof. Pietro Roversi |
| **Chemical, enzymes and other reagents** | | |
| Bis[sulsosuccinimidyl]suberate | Thermo Fisher | 26173 |
| sulfosuccinimidyl 4,4′-azipentanoate | Thermo Fisher | 21580 |
| Trypsin | Thermo Fisher | 90057 |
| AspN | Promega | V1621 |
| Chymotrypsin | Promega | V1061 |
| GluC | Promega | V1651 |
| NuPAGE™ Novex 4–12% Bis-Tris | Thermo Fisher | NP0335BOX |
| Dithiothreitol | Merck | 1114740005 |
| Iodoacetamide | SIGMA | I1149 |
| Ammonium bicarbonate | SIGMA | A6141 |
| TFA | SIGMA | T6508 |
| Formic acid | Fluka | 94318 |
| Acetonitrile | Riedel-de Haen | 34967 |

**Reagents and Tools table** (continued)

| Reagent/resource | Reference or source | Identifier or catalogue number |
|---|---|---|
| **Equipment** | | |
| Superdex Peptide 3.2/300 | GE Healthcare | GE29-0362-31 |
| Shimadzu HPLC | Shimadzu | |
| EASY-Spray™ LC column | Thermo Fisher | ES803 |
| Dionex Ultimate 3000 RSLCnano | Thermo Fisher | |
| Orbitrap Fusion Lumos Tribrid | Thermo Fisher | |

## Methods and Protocols

### Sample preparation, crosslinking and digestion with trypsin

The seven standard proteins catalase, myoglobin, cytochrome C, lysozyme, creatine kinase, HSA and conalbumin were resuspended in BS$^3$ crosslinking buffer (20 mM HEPES, 20 mM NaCl, 5 mM MgCl$_2$, pH 7.8) to a final concentration of 1 mg/ml. Crosslinker was added to a 1:1 ($w/w$) protein to crosslinker ratio and samples incubated for 2 h on ice. Crosslinking reaction was quenched with excess ammonium bicarbonate (ABC) for 1 h at room temperature (RT). The seven crosslinked proteins were loaded on NuPAGE™ 4–12% Bis-Tris protein gels to isolate the monomeric band of each protein that was then extracted, and in-gel digested with trypsin (Shevchenko *et al*, 2006). After peptide extraction from the gel, peptides from each protein were mixed in a 1:1 weight ratio to a final amount of 200 μg divided into four parts and desalted using C18-StageTips (Rappsilber *et al*, 2003).

For the OCCM complex, pUC19-ARS1 beads were used to assemble the OCCM complex as described elsewhere (Evrin *et al*, 2009). Crosslinking was performed on beads. BS$^3$ was added to 200 μg of the OCCM complex to a 1:8,100 protein to crosslinker molar ratio. The sample was incubated for 2 h on ice, and the crosslinking reaction was quenched with excess ABC for 1 h at RT. The sample was transferred into 8 M urea, reduced with dithiothreitol (DTT), alkylated with iodoacetamide (IAA) and diluted with ABC 50 mM to a final concentration of 2 M urea. Trypsin was added to a protease-to-substrate ratio of 1:50, and the sample was incubated ON at 37°C. Reaction was stopped with 10% ($v/v$) trifluoroacetic acid (TFA), and the samples were divided into four parts and desalted using StageTips.

The 26S proteasome was isolated from *S. cerevisiae* by affinity purification using the 3× FLAG-tagged subunit Rpn11 as described elsewhere (Sakata *et al*, 2011). For the crosslinking, the 26S proteasome buffer was exchanged to BS$^3$ crosslinking buffer using 30 kDa molecular weight cut-off (MWCO) filters (Millipore). 200 μg of the 26S proteasome was crosslinked with BS$^3$. BS3 was added to a 1:1 ($w/w$) protein to crosslinker ratio. Samples were incubated for 2 h on ice, and the crosslinking reaction was quenched with excess ABC for 1 h at room temperature (RT). The sample was dried using a vacuum concentrator and resuspended in 6 M urea/2 M thiourea for subsequent *in-solution* digestion. Sample was reduced with 2.5 mM DTT for 15 min at 50°C, then alkylated with 5 mM IAA at RT in the dark and diluted with ABC 50 mM to a final concentration of 1 M. Trypsin was added at an enzyme-to-substrate mass ratio of 1:50, and the sample was incubated ON at 37°C. Reaction was stopped with 10% ($v/v$) TFA, and the samples were divided into four parts and desalted using C18-StageTips.

K562 cells (DSMZ, Cat# ACC-10, negatively tested for mycoplasma) were grown in T175 flasks at 37°C in humidified 5% ($v/v$) CO$_2$ incubators in RPMI 1640 media supplemented with 10% ($v/v$) foetal bovine serum (FBS) + 2 mM glutamine. $3 \times 10^8$ cells were harvested by centrifugation ($180 \times g$) and washed three times with ice-cold PBS. Cells were lysed in lysis buffer (100 mM HEPES pH 7.2, 100 mM KCl, 20 mM NaCl, 3 mM MgCl$_2$, 1 mM EDTA, 10% ($v/v$) glycerol, 1 mM DTT, 10 μg/ml DNAse1, complete EDTA-free protease inhibitor cocktail, 1 mM PMSF) using a douncer at 4°C. Lysate was cleared of debris by centrifugation at $100,000 \times g$ for 45 min. Native protein complexes were further concentrated by spin filtration using a 100,000 Da cut-off Amicon Ultra centrifugal filter unit (approx. 30 min). For protein co-elution analysis, 100 μl of concentrated lysate (approximately 30 mg/ml) was separated by a Biosep SEC-S4000 ($7.8 \times 600$) size exclusion column on an Åkta Purifier (HPLC) system running at 0.25 ml/min 100 mM HEPES pH 7.2, 100 mM KCl, 20 mM NaCl, 3 mM MgCl$_2$, 1 mM EDTA and 10% glycerol. 500 μl fractions were collected. Protein fractions were crosslinked at < 1 mg/ml (quantified by Bradford) and with 2:1 $w/w$ ratio of BS$^3$ to protein. Samples were *in-solution* digested with trypsin as described above. Samples were divided into four parts and desalted using C18-StageTips.

C3b monomer and dimer were labelled as described elsewhere (Chen *et al*, 2016c) and in-gel digested with trypsin. Samples were divided into four parts and desalted using C18-StageTips.

UGGT was crosslinked using sulfo-SDA using eight different proteins to crosslinker ratios [1:0.13, 1:0.19, 1:0.25, 1:0.38, 1:0.5, 1:0.75, 1:1 and 1:1.5 ($w/w$)]. Crosslinking was carried out in two stages: firstly, sulfo-SDA, dissolved in SDA crosslinking buffer (25 μl, 20 mM HEPES-OH, 20 mM NaCl, 5 mM MgCl$_2$, pH 7.8), was added to target protein (25 μg, 1 μg/μl) and left to react in the dark for 50 min at room temperature. The diazirine group was then photo-activated using UV irradiation, at 365 nm, from a UVP CL-1000 UV Crosslinker (UVP Inc.). Samples were spread onto the inside of Eppendorf tube lids by pipetting (covering the entire surface of the inner lid), placed on ice at a distance of 5 cm from the tubes and irradiated for 20 min. The reaction mixtures from the eight conditions corresponding to each experiment were combined and quenched with the addition of saturated ABC (13 μl). Sample was dried in a vacuum concentrator, and 200 μg of protein was *in-solution* digested with trypsin as described above. Sample was divided into four parts and desalted using C18-StageTips.

For all models, one part of the sample digested with trypsin was fractionated by SEC and the remaining three parts were sequentially digested with AspN, chymotrypsin and GluC, respectively.

### Parallel digestion

Parallel digestion with AspN, chymotrypsin and GluC of the seven standard proteins was performed as described for trypsin. After isolating the monomeric bands of each one of the seven standard proteins, those were in-gel digested in parallel with AspN, chymotrypsin and GluC. After peptide extraction from the gel, peptides from each protein digested with the same protease were mixed in a 1:1 weight ratio to a final amount of 200 μg resulting in four samples of the protein standards, one digested with AspN, one digested with chymotrypsin and one digested with AspN. Proteases were added as follows: GluC and chymotrypsin were added to a protease-to-substrate (w/w) ratio of 1:50 and incubated overnight (ON) at 37°C and RT, respectively; AspN was added to a protease-to-substrate ratio (w/w) of 1:100 and incubated ON at 37°C. After parallel digestion, samples were fractioned by SEC and analysed by LC-MS/MS.

### Sequential digestion

After desalting of tryptic peptides, sequential digestion was performed as follows:

1  Ressuspend tryptic peptides in ABC 50 mM.
2  For sequential digestion with AspN:
    2.1    Add AspN to a final protease:protein ratio of 1:100 (w/w).
    2.2    Incubate ON at 37°C.
    2.3    Acidify samples using 10% TFA.
    2.4    Reduce sample volume to 50 μl by evaporation using a vacuum concentrator.
    2.5    Fractionate by SEC.
3  For sequential digestion with chymotrypsin:
    3.1    Add chymotrypsin to a final protease:protein ratio of 1:50 (w/w).
    3.2    Incubate ON at RT.
    3.3    Acidify samples using 10% TFA.
    3.4    Reduce sample volume to 50 μl by evaporation using a vacuum concentrator.
    3.5    Fractionate by SEC.
4  For sequential digestion with GluC:
    4.1    Add GluC to a final protease:protein ratio of 1:50 (w/w).
    4.2    Incubate ON at 37°C.
    4.3    Acidify samples using 10% TFA.
    4.4    Reduce sample volume to 50 μl by evaporation using a vacuum concentrator.
    4.5    Fractionate by SEC.

### Fractionation of peptides by size exclusion chromatography

Size exclusion chromatography (SEC) of crosslinked peptides was performed as described elsewhere (Leitner et al, 2013). 50 μg of peptides were fractionated in a Shimadzu HPLC system using a Superdex Peptide 3.2/300 (GE Healthcare) at a flow rate of 50 μl/min using SEC buffer (30% (v/v) ACN, 0.1% (v/v) TFA) as mobile phase. Separation was monitored by UV absorption at 215 and 280 nm. Fractions were collected every 2 min over one column volume. The three high-MW fractions were dried, resuspended in 0.1% (v/v) TFA and analysed by LC-MS/MS. All samples in this work, including all the samples used in our proof of principle experiments—replicates, parallel and sequential digestions—of the seven standard proteins, were fractionated by SEC as described in our workflow in Fig 1A.

### In vitro pre-RC assay

The in vitro pre-RC assay was performed as described (Evrin et al, 2009; Fernández-Cid et al, 2013). Briefly, ORC (40 nM), Cdc6 (80 nM), Cdt1 (40 nM), MCM2-7 (40 nM) or MCM2-7 ΔC2 (40 nM) were incubated with 6 nM of DNA containing the ARS1 DNA sequence in 50 μl of pre-RC buffer (50 mM HEPES pH 7.5, 100 mM potassium glutamate, 10 mM magnesium acetate, 50 μM zinc acetate, 3 mM ATP, 5 mM DTT, 50 μM EDTA, 0.1% Triton-X100, 5% glycerol). After 20 min at 24°C, the reactions were washed three times with low salt buffer (pre-RC buffer) or high salt buffer (pre-RC buffer plus 300 mM NaCl) before digestion with 1 U of DNaseI in pre-RC buffer plus $CaCl_2$.

### Yeast lethality assay

Yeast strain AS499 (MATa. bar1Δ, leu2-3,-112, ura3-52, his3-Δ200, trp1-Δ-63, ade2-1 lys2-801, pep4) was transformed with pESC-LEU-MCM2-MCM7, pESC-TRP-MCM6-MCM4 and pESC-URA-HA-MCM3-MCM5 (wild type, YC119) or pESC-LEU-MCM2 (Δ848–868)-MCM7 (MCM2-7ΔC2, YC388). Yeast strains YC119 and YC388 were plated on a dropout synthetic complete (SC) medium and incubated at 30°C for 48 h. Cells were grown in suspension to $10^7$ cells/ml. Three microlitres of a fivefold serial dilution was spotted onto selective plates containing either 2% galactose or glucose. Plates were incubated at 30°C for 3–5 days.

### LC-MS/MS

Samples were analysed using a Thermo Scientific Dionex Ultimate 3000 RSLCnano system coupled to a Thermo Scientific Orbitrap Fusion Lumos Tribrid mass spectrometer equipped with an EASY-Spray source. Mobile phase A consisted of 0.1% (v/v) FA in water and mobile phase B consisted of 80% (v/v) ACN and 0.1% (v/v) FA in water. Peptides were loaded into a 50 cm EASY-Spray column operated at a temperature of 45°C at a flow rate of 300 nl/min and separated at 300 nl/min using the following gradient: 2% mobile phase B (0–11 min); 2–40% mobile phase B (11–150 min); 40–95% mobile phase B (150–161 min); 95% mobile phase B (161–166 min); 95–2% mobile phase B (166–185 min).

MS data were acquired using a "high-high" acquisition method using the Orbitrap to detect both MS and MS/MS scans. The instrument was operated in a data-dependent mode with a cycle time of 3 s. MS1 scans were acquired at a resolution of 120,000 using a scan range from 300 to 1,700 $m/z$ and AGC target of $2.5 \times 10^5$ with a maximum injection time of 50 ms. The monoisotopic peak determination (MIPS) was activated, and only precursors with charge states between 3 and 8 with an intensity threshold higher than $5.0 \times 10^4$ were selected for fragmentation. Selected precursors were fragmented by HCD using a collision energy setting of 30%. MS2 spectra were acquired at a resolution of 15,000 and AGC of $10^4$ with a maximum injection time of 35 ms. Dynamic exclusion was set to 60 s after one count.

### Data analysis

Thermo raw data were pre-processed using MaxQuant (v 1.5.7.4) to extract the peak list files (APL format). Partial processing in MaxQuant was performed until step 5 with the parameters set to default with the exception of the "FTMS top peaks per Da interval" which was set to 100 and no FTMS de-isotoping was allowed. apl files were uploaded to xiSEARCH for identification of crosslinked peptides (xiSEARCH software is available from https://rappsilbe

rlab.org/downloads/and the code is available from https://github.com/Rappsilber-Laboratory/xiSEARCH). For the crosslinking search, the parameters used were as follows: MS accuracy, 6 ppm; MS/MS accuracy, 20 ppm; enzyme, trypsin, trypsin + AspN, trypsin + chymotrypsin or trypsin + Gluc depending on the sample digestion conditions; missed cleavages allowed, 4. For the BS$^3$ crosslinked samples, carbamidomethylation of cysteines was set as a fixed modification and oxidation of methionine, the hydrolysed crosslinker (BS-OH: 156.0786 Da) and the amidated crosslinker (BS3-NH2: 155.0964 Da) were set as variable modifications. Reaction specificity for BS$^3$ (mass modification: 138.0681 Da) was assumed to be with Lysine, Serine, Threonine and Tyrosine or the protein N-terminus. For SDA crosslinked samples, carbamidomethylation of cysteines, oxidation of methionine and the crosslinker alone (mass modification: 109.0396 Da), hydrolysed (SDA-OH: 100.0524 Da) or crosslinker loop (SDA-loop: SDA crosslink within a peptide that is also crosslinked to a separate peptide, 82.0419 Da) were set as variable modifications. Reaction specificity for SDA was assumed to be with lysine, serine, threonine and tyrosine or the protein N-terminus on one end of the spacer and with all residues on the other end. Apl files were searched against the following databases: for the Standard Protein Mix, a database was built containing the sequences corresponding to the crystal structures of the standard proteins used in the mix (PDB accession: 3j7u, 5d5r, 3nbs, 1dpx, 2crk, 1ao6 and 1ova); for C3b, we used the FASTA corresponding to the UniProt accession P01024; for the UGGT, the UniProt sequence of the protein was used (G0SB58); for the OCCM complex, a database containing all the 14 OCCM subunits was used; and for the 26S proteasome, a linear search of the data was first performed using a complete *S. cerevisiae* database (UniProt, release-2016_11) and the proteins that were present at < 1% of the most abundant protein (as judged by iBAQ values) were excluded. Our rationale for excluding the least abundant proteins from crosslink searches was that crosslinks tend to be less easily identified than linear peptides and thus will be detected only in the more abundant proteins. Although likely not contributing detectable crosslinks, less abundant proteins will nevertheless contribute random matches and thus reduce the sensitivity of the search. For the human cytosol, linear searches were performed for each one of the initial SEC fractions using the entire human database (UniProt, release-2016_02). Proteins that were present at < 1% (SEC fraction 1) or 0.5% (SEC fractions 2–7) of the most abundant protein (as judged by iBAQ values) were excluded. The different threshold was used to result in databases of about equal size. FDR was estimated on a 5% residue level, including only unique PSMs and boosting results, using xiFDR (Fischer & Rappsilber, 2017).

Cleavage site protection was calculated by dividing the mean of available miss-cleavages for the second protease in the sequential digestion datasets by the mean of the potential miss-cleavages in the trypsin dataset.

For software comparison, the 26S proteasome was searched using xiSEARCH (version 1.6.731), Kojak (version 1.5.5) and pLink 2 (version 2.3.2) using the same parameters as described before with minor alterations: Searches were performed using two missed cleavages, MS accuracy 3 ppm and MS/MS accuracy 20 ppm. For xiSEARCH, we used missing monoisotopic peaks 3. For xiSEARCH, Kojak and pLink 2, we gave the same preference for linkage sites in K, S, T and Y.

## Data availability

The datasets and computer code produced in this study are available in the following databases:

- Raw-files and identifications: ProteomeXchange Consortium (http://proteomecentral.proteomexchange.org) via the PRIDE partner repository (Vizcaíno *et al*, 2014) at: https://www.ebi.ac.uk/pride/archive/projects/PXD008550.
- The source code of xiSEARCH is available at: https://github.com/Rappsilber-Laboratory/XiSearch
- The source code of xiFDR is available at: https://github.com/Rappsilber-Laboratory/xiFDR

**Expanded View** for this article is available online.

## Acknowledgements

We thank Swantje Lenz for the technical support and helpful discussions and Petra Ryl for sample preparation support. We thank Pietro Roversi and CASP12 organisers for supplying UGGT sample and Alberto Riera for contributing OCCM sample. We thank Michael Hoopmann for his support with pairing kojak and peptideProphet. M.L.M. was supported by the International Post-Doc Initiative—IPODI, co-funded by the European Union. This work was supported by the Einstein Foundation, the DFG (RA 2365/4-1), and the Wellcome Trust through a Senior Research Fellowship to J.R. (103139), an Investigator Award to C.S. (107903/Z/15/Z) and a multi-user equipment grant to J.R. (108504), the Biotechnology and Biological Sciences Research Council UK to C.S. (BB/N000323/1) and the Medical Research Council UK to C.S. (MC_U120085811). The Wellcome Centre for Cell Biology is supported by core funding from the Wellcome Trust (203149).

## Author contribution

MLM, LF and JR developed the study; MLM, ZAC, FJO'R, AB, MB and MRE performed sample preparation; MLM performed LC-MS analysis; MLM, ZAC, FJO'R, AB, TD, MB, CS, SG and MB-S performed data analysis; LF developed xiSEARCH; CWC and MG provided critical support in data analysis; MB, CS, MRE and WB contributed with data interpretation; and MLM, FJO'R, MB-S, LF and JR wrote the article with critical input from all authors.

## Conflict of interest

The authors declare that they have no conflict of interest.

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
