## [Review Process File · Molecular Systems Biology]

An integrated workflow for crosslinking mass spectrometry to identify dynamic and thus druggable protein-protein interactions

Marta L. Mendes, Lutz Fischer, Zhuo A. Chen, Marta Barbon, Francis J. O'Reilly, Sven Giese, Michael Bohlke-Schneider, Adam Belsom, Therese Dau, Colin W. Combe, Martin Graham, Markus R. Eisele, Wolfgang Baumeister, Christian Speck and Juri Rappsilber.

Review timeline:

Submission date:	9 th May 2019
Editorial Decision:	25 th June 2019
Revision received:	19 th August 2019
Accepted:	21 st August 2019

Editor: Maria Polychronidou

Transaction Report:

1st Editorial Decision

25th June 2019

Thank you again for submitting your work to Molecular Systems Biology. We have now heard back from the three referees who agreed to evaluate your study. As you will see below, the reviewers raise a series of concerns, which we would ask you to address in a revision.

As you will see below, reviewer #3 is somewhat less enthusiastic regarding the overall conceptual novelty of the study. However, we do not think that the remarks of reviewer #3 on the novelty disqualify the manuscript, since they are not shared by the other reviewers. Moreover, reviewer #3 does appreciate that, as reviewers #1 and #3 also point out, the presented workflow will likely be useful for the community.

I think that overall the reviewers' recommendations are rather clear and there is therefore no need to repeat the points listed below. All issues raised by the reviewers need to be satisfactorily addressed. Please feel free to contact me in case you would like to discuss in further detail any of the issues raised.

REFERE REPORTS

Reviewer #1:

The manuscript outlines a powerful workflow for LC-MS analyses of crosslinked samples. This topic receives currently a lot of attention within the proteome community. The main theme of the work is the application of sequential digestion of crosslinked protein complexes with various endoproteinases that ultimately led to better coverage of crosslinked peptides in the subsequent database search. The results are compelling also in quantitative manner. In addition, the manuscript

finally introduces (officially) the software tool Xi for the reliable identification of cross-linked peptides. In my opinion, Xi is a very good software especially when it comes to FDR estimation. From the authors comparison Xi seems to outperform other crosslink software tools. Given the relevance of crosslink analyses and the fundamental work of the Rappsilber lab and the here presented data I suggest to consider publication.

One cavity of this work is that the authors used mainly relative "simple" protein complexes to exemplify their workflow although they performed crosslinking and sequential digestion on yeast cells. Reading the Materials and Methods the authors write that they created a database of crosslinked yeast proteins with the top 1 % of proteins according to the highest iBAQ values of proteins. How many were these? It is important to know if the sequential digestion workflow can be really applied on large crosslinked proteomes, for example the entire yeast database. It is clear that many endoproteases are specific for one or two residues while chymotrypsin has a broad spectrum. Can the software cope with this with a still valuable FDR when searching the entire database?

I do not completely understand the comparison and the results of the different search engines in Figure 3b. More than 2300 (in words two thousand three hundred) unique crosslink pairs have been identified with Xi in cross-linked 26S proteasome. The other software also identified quite a very high number of cross-links as well. This is truly amazing! But: Figure 2 plots unique crosslinks on the structure of the 26S proteasome. These are definitely not 2300 and it already seems that the identified crosslinks connect every available lysine residue? I would like that the authors clarify this. Most search engine still give 1 MSMS spectra for a crosslink especially upon raising sample complexity. It would be helpful when the authors differentiate in their data analyses between unique crosslinks and spectra matches and compare Xi in more detail when they compare it with the other software.

Reviewer #2:

In the paper by Mendes et al the authors describe their attempts at improving crosslinking mass spectrometry studies by utilizing multiple proteases. Related work has been described before by e.g. Leitner et al MCP 2012. The added feature the authors apply here is that instead of applying the proteases in isolation and analyzing the peptides, the authors combine multiple enzymes in a 'single pot' either in parallel or sequentially. Given that this is a new concept and the paper is well written and explanatory, I would deem the manuscript worthy of publication. However, there are a number of major concerns that should be addressed.

Major issues

[1] I find the title a bit misleading and at least not fully correct. The paper describes the effect of using various proteases on the total number of identifications and this should be reflected there.

[2] I applaud the authors for going into quite some detail about the properties of the peptides identified, however the manuscript lacks any description of the fragmentation spectra supporting the identifications. The main problem the authors are addressing lies there - the peptides become shorter and can be better fragmented. It would be insightful to see plots describing sequence coverage of the individual peptides in the pairs, improvements in overall scores, improvements in intensity of the fragment peaks and very likely many more metrics giving hints towards increased data quality.

[3] The quantitative aspect in its current form is hardly described with Supplementary Fig 5b the only panel on this topic - and this simply describes dynamic range. The protein C3b is quantified, but the resulting information is never used. I would argue that in its current form the quantitation information should be removed. Alternatively, it would be insightful if the authors can demonstrate that with the protease trick reproducible quantities can be extracted (something which I hope will be the case) and whether the various combinations of proteases give rise to identical quantities for the same lysine-lysine pairs (i.e. not peptide-pairs), which will be much harder due to experimental variation.

[3] The very nice find of over-length crosslinks mapping to the base of the proteasome is now buried in supplementary fig 7. Given the low number of figures, this would be very well placed in the main text.

[4] The sequential digestion is a happy accident being the best of the two tested approaches and it is a nice find of the authors. However, the added complexity of utilizing multiple enzymes leads one to suspect more available individual species. It would be insightful if the authors can provide an

overview of available precursor isotopes from the trypsin and combination digestions. Additionally, could the authors at the very least speculate if there is an increase whether longer analysis times would be in order.

[5] The limited overlap between the study of Liu et al and this study can be more readily explained in that the authors here took only high molecular weight SEC fractions of the cytosolic lysate. The paper by Liu et used a full lysate. The authors should point this out.

Minor issues

[1] Page 2, XlinkX is not limited to DSSO only. It never has been in the online version and also not in its current iteration within Proteome Discoverer.

[2] Figure 1b, it would be insightful if the authors provide the number of replicates used for the bar charts - best within the chart.

Reviewer #3:

An integrated workflow for crosslinking mass spectrometry

Marta L. Mendes#, Lutz Fischer#, Zhuo A. Chen, Marta Barbon, Francis J. O'Reilly, Sven Giese, Michael Bohlke-Schneider, Adam Belsom, Therese Dau, Colin W. Combe, Martin Graham, Markus R. Eisele, Wolfgang Baumeister, Christian Speck, Juri Rappsilber

The authors describe a method for the consecutive digestion of cross-linked protein-complexes using different proteases and report an updated version of their database search engine 'Xi'. The authors use the properties of different proteases in combination and compare tryptic peptides to peptides derived from parallel digestion and sequential digestion of trypsin with either AspN, GluC or chymotrypsin. The number of unique identifications increases by combining peptides obtained from trypsin digest with peptides from the sequential digest with trypsin and either AspN, GluC or chymotrypsin. The bioinformatic analysis of the mass spectrometric data from parallel and sequential digestion is achieved by a newly developed feature of their search algorithm Xi. I appreciate the demonstrated benefit of sequential digestion for crosslinking analysis, however, the technical advantage of consecutive digests has been published before as well as has been the Xi software for non-functionalized crosslinkers. Neither the technical developments nor the interpretation of the crosslink information on structural models provide sufficient novelty to justify publication of this manuscript in *Molecular Systems Biology*. Besides this, I am wondering whether a report on a technical development in the crosslinking field fits the scope of *Molecular Systems Biology*. As this work is a valuable contribution to the development of crosslinking workflows, I recommend transferring this manuscript to a more specialized analytical/proteomics journal.

General Comments:

1. The finding of the Authors was anticipated since various research articles including their own work describe similar approaches:

- a. Dau T, Gupta K, Berger I, Rappsilber J. Sequential digestion with Trypsin and Elastase in cross-linking/mass spectrometry. *Analytical Chemistry* 2019 91 (7): 4472-4478
- b. Wiśniewski J and Mann M. Consecutive Proteolytic Digestion in an Enzyme Reactor Increases Depth of Proteomic and Phosphoproteomic Analysis. *Analytical Chemistry* 2012 84 (6): 2631-2637
- c. Gilmore JM, Kettenbach AN, Gerber SA. Increasing phosphoproteomic coverage through sequential digestion by complementary proteases. *Anal Bioanal Chem.* 2011;402(2):711-720.
- d. Guo X, Trudgian DC, Lemoff A, Yadavalli S, Mirzaei H. Confetti: a multiprotease map of the HeLa proteome for comprehensive proteomics. *Mol Cell Proteomics.* 2014;13(6):1573-1584.
- e. Larsen MR1, Højrup P, Roepstorff P. Characterization of gel-separated glycoproteins using two-step proteolytic digestion combined with sequential microcolumns and mass spectrometry. *Mol Cell Proteomics.* 2005 Feb;4(2):107-19.
- f. And others

We thank the reviewers for their comments and below we address their concerns, inserting our comments whenever we felt there was a need for clarification.

Reviewer #1:

The manuscript outlines a powerful workflow for LC-MS analyses of crosslinked samples. This topic receives currently a lot of attention within the proteome community. The main theme of the work is the application of sequential digestion of crosslinked protein complexes with various endoproteases that ultimately led to better coverage of crosslinked peptides in the subsequent database search. The results are compelling also in quantitative manner. In addition, the manuscript finally introduces (officially) the software tool Xi for the reliable identification of cross-linked peptides. In my opinion, Xi is a very good software especially when it comes to FDR estimation. From the authors comparison Xi seems to outperform other crosslink software tools. Given the relevance of crosslink analyses and the fundamental work of the Rappsilber lab and the here presented data I suggest to consider publication.

One cavity of this work is that the authors used mainly relative "simple" protein complexes to exemplify their workflow although they performed crosslinking and sequential digestion on yeast cells.

We thank the reviewer for sharing our excitement for the presented work. We would like to clarify that in addition to "relative "simple" protein complexes" we analysed yeast proteasomal preparations that nevertheless contained over 500 proteins (637) and the cytosolic fraction of human K562 cells, detecting 2847 proteins. Note that we did not find crosslinks for all proteins that we identified, which likely is due to the low abundance of crosslinks compared to linear peptides, making it easier to identify a protein than to see crosslinks for it.

Reading the Materials and Methods the authors write that they created a database of crosslinked yeast proteins with the top 1 % of proteins according to the highest iBAQ values of proteins. How many were these?

We suspect the reviewer refers here to our analysis of human cytosol samples. We excluded proteins that were present at lower abundance than 1% of the most abundant protein (as judged by iBAQ values) in each SEC fraction. As the composition of fractions varies so does the size and composition of the respectively used database. To name fraction 1 as an example, 1087 proteins were identified in this fraction. 433 proteins had an IBAC higher than 1% of the most abundant protein in that fraction and were included in the database, while 654 proteins had a smaller IBAC and were excluded. This follows the logic that crosslinks tend to be less abundant than linear peptides and are likely identified only for the more abundant proteins. For other fractions, since the number of proteins above this cut-off was so low we decided to increase the cut-off to 0.5% iBAQ leading to the inclusion of around 400 proteins and excluding 600.

The same 1% cutoff was applied also during the analysis of our yeast proteasomal fraction, leading to 151 proteins while excluding 486 proteins from the search database.

Note that the motivation for this is not a simplification of search space for technical reasons (i.e. the ability of Xi to work with a database of thousands of proteins) but the reduction of random matches. It is advisable to focus the search space as much as possible onto proteins with detectable crosslinks, although these proteins are of course not known accurately beforehand. We use abundance as an approximation. We clarified the description in the manuscript: "[...] proteins that were present at less than 1% of the most abundant protein (as judged by iBAQ values) were excluded". (page 23, paragraph 1)

It is important to know if the sequential digestion workflow can be really applied on large crosslinked proteomes, for example the entire yeast database.

We applied the workflow to human cytosolic fractions. Xi can search the entire human proteome. Nevertheless, the ideal database contains only entries that actually are in the sample and are detectable. When choosing human database for a human sample one already applies this logic to some extent. Reducing the database further from all theoretical human proteins to those identified in a given sample continues this logic. Adding then an abundance threshold exploits the knowledge that cross-linked peptides are less abundant than linear ones. Reducing the database in this way enhances the chances of matching the right sequences, i.e. reduces background and maximises results.

We did now also search a combination of yeast and *E. coli* databases. For results see next response.

It is clear that many endoproteinases are specific for one or two residues while chymotrypsin has a broad spectrum. Can the software cope with this with a still valuable FDR when searching the entire database?

We understand this question to fall into two parts. Firstly, can xiSEARCH cope with the increased search space resulting from the increased cleavage sites of sequential digestion? For example, in the case of Trypsin-Chymotrypsin sequential digestion we searched for all analyses (including human cytosolic fractions) with a specificity of K, R, F, L, W, and Y. So, we think the answer is yes. Secondly, does our FDR approach still hold? To test this we searched our yeast proteasome data obtained by Trypsin/Chymotrypsin sequential digestion against a database containing all yeast and all *E. coli* SwissProt protein sequences (6721 and 4350, respectively). Note that this search revealed only 181 crosslinks (target-target) compared to the 678 (Appendix Figure S4) as we searched through a much larger database (more background) but also because we changed the search settings (one missed cleavage, no modifications) to minimise computational costs. Importantly, of these 181 crosslinks only one (0.55%) involved a peptide from *E. coli* and thus was obviously wrong. This demonstrates that our FDR approach is not invalidated, neither by database size nor by sequential digestion.

I do not completely understand the comparison and the results of the different search engines in Figure 3b. More than 2300 (in words two thousand three hundred) unique crosslink pairs have been identified with Xi in cross-linked 26S proteasome. The other software also identified quite a very high number of crops-links as well. This is truly amazing! But: Figure 2 plots unique crosslinks on the structure of the 26S proteasome. These are definitely not 2300

In fig 2 only the links fitting the published structure were represented. In that search, which differs from the search in the comparison work (we prioritised Lysine as linkage site), we identified 2098 links. Some of these are in other proteins than the proteasome and some are within the proteasome but not in regions of the proteasome covered by the solved structure. 1249 fell within protein regions covered by the published structure. These 1249 are displayed in Fig. 2.

and it already seems that the identified crosslinks connect every available lysine residue? I would like that the authors clarify this.

Rpn1-3, Rpn5–14, RPT1-6, protein α 1-7 and β 1-7 contain 897 K, 877 S, 657 T and 428 Y residues. If all were linked to all this would result in 4088370 links. If one were to assume only K-K links this would give rise to 402753 links. We observed 2098 links. We certainly do not see all possible combinations of linkable residues. Considering this comment differently, of the 897 K in the sequences of the proteins we see 656 involved in crosslinks and of the 2879 linkable residues we see 1006 involved in crosslinks.

Most search engine still give 1 MSMS spectra for a crosslink especially upon raising sample complexity. xiSEARCH optionally provides a csv file with annotations for each reported spectrum-match. However, manually looking through many thousand matches appears to us not meaningful. For manual interrogation our lab provides xiVIEW (<https://xiVIEW.org>) to which any interested scientist can upload our (and other) data downloaded from PRIDE.

It would be helpful when the authors differentiate in their data analyses between unique crosslinks and spectra matches and compare Xi in more detail when they compare it with the other software.

We present a comparison of unique residue pairs, the current outcome of crosslink experiments. This appears to us the minimal basis of a comparison.

We would prefer not to go into larger details and through this emphasize more the software comparison. While we understand the desire of users for software comparisons, such an analysis should really not be done by a single developer lab. Each group knows their own software best and will be biased for their own software - if only subconsciously or through choice of evaluation metric. Therefore these comparisons are of limited use - unless done in a way that minimises the influence of an individual developer and that ideally involves the authors of all software packages that are being evaluated.

Reviewer #2:

In the paper by Mendes et al the authors describe their attempts at improving crosslinking mass spectrometry studies by utilizing multiple proteases. Related work has been described before by e.g. Leitner et al MCP 2012. The added feature the authors apply here is that instead of applying the proteases in isolation and analyzing the peptides, the authors combine multiple enzymes in a 'single pot' either in parallel or sequentially. Given that this is a new concept and the paper is well written and explanatory, I would deem the manuscript worthy of publication. However, there a number of major concerns that should be addressed.

Major issues

[1] I find the title a bit misleading and at least not fully correct. The paper describes the effect of using various proteases on the total number of identifications and this should be reflected there.

The paper presents a workflow involving sequential digestion and the xi software in an integrated workflow that additionally features SEC as a technology. The current title "An integrated workflow for crosslinking mass spectrometry" appears still appropriate to us but could be expanded to "An integrated and compact workflow for crosslinking mass spectrometry, including sequential digestion and the Xi search engine" to name key novel elements of the workflow. We fear that a title that tries to be comprehensive also becomes rather long, however, and would suggest sticking with the current version.

[2] I applaud the authors for going into quite some detail about the properties of the peptides identified, however the manuscript lacks any description of the fragmentation spectra supporting the identifications. The main problem the authors are addressing lies there - the peptides become shorter and can be better fragmented. It would be insightful to see plots describing sequence coverage of the individual peptides in the pairs, improvements in overall scores, improvements in intensity of the fragment peaks and very likely many more metrics giving hints towards increased data quality.

We actually are less convinced than the reviewer as to how many insights can be gleaned from further analyses of the data. It is not actually clear what should be compared here. The second digestion step shortens long tryptic peptides and brings them into a mass range that is more observable by mass spectrometry. This is an MS1 phenomenon that we describe in detail in our manuscript. MS2 aspects as suggested by the review may or may not play a role. What would the MS2 of sequentially digested peptides be compared to? To the original tryptic peptide that was likely not detected? To tryptic peptides of same size that thus are often indistinguishable as many of our detected sequentially digested peptides possess a

tryptic C-terminus? Importantly, we provide public access to our data and scientist with a better set of ideas than us can therefore pick up on this thread if interested.

[3] The quantitative aspect in its current form is hardly described with Supplementary Fig 5b the only panel on this topic - and this simply describes dynamic range. The protein C3b is quantified, but the resulting information is never used. I would argue that in its current form the quantitation information should be removed.

C3b dimerization was studied to show the compatibility of our workflow with QCLMS. This goal was achieved by identifying and quantifying additional residue pairs supporting a bottom to bottom interaction. The quantitative C3b study is described in the main text by its own paragraph that covers the technical aspects of this experiment and biological conclusions, backed by data in the entire figure mentioned by the reviewer (now Appendix Figure 2) and not only its panel b.

Alternatively, it would be insightful if the authors can demonstrate that with the protease trick reproducible quantities can be extracted (something which I hope will be the case) and whether the various combinations of proteases give rise to identical quantities for the same lysine-lysine pairs (i.e. not peptide-pairs), which will be much harder due to experimental variation.

We compared the quantitation outcome for crosslinks based on trypsin data versus that of sequential digestion and prepared a new figure panel for Appendix Figure 2 (see below).

Figure: Comparisons of ratios observed in different sequential digests compared to Trypsin only digest. Green highlighted are links detected as dimer specific

[3] The very nice find of over-length crosslinks mapping to the base of the proteasome is now buried in supplementary fig 7. Given the low number of figures, this would be very well placed in the main text.

We have moved this data into the main text figures (Figure 3).

[4] The sequential digestion is a happy accident being the best of the two tested approaches and it is a nice find of the authors. However, the added complexity of utilizing multiple enzymes leads one to suspect more

available individual species. It would be insightful if the authors can provide an overview of available precursor isotopes from the trypsin and combination digestions.

We followed the suggestion of the reviewer and did not find an increase in complexity, at least when relying on MaxQuant's feature finding algorithm (see figures below). This is surprising and we have no explanation as of now.

Additionally, could the authors at the very least speculate if there is an increase whether longer analysis times would be in order.

We would expect that a longer gradient leads to more links regardless of a change in complexity. At some point this is not true any longer as peak broadening reduces intensities, of course.

[5] The limited overlap between the study of Liu et al and this study can be more readily explained in that the authors here took only high molecular weight SEC fractions of the cytosolic lysate. The paper by Liu et used a full lysate. The authors should point this out.

We thank the reviewer for pointing this out and have amended our manuscript accordingly: “[...] this will be influenced by factors such as the different starting material and the different analytical strategies.” (page 8, paragraph 1)

Minor issues

[1] Page 2, XlinkX is not limited to DSSO only. It never has been in the online version and also not in its current iteration within Proteome Discoverer.

We have removed this section.

[2] Figure 1b, it would be insightful if the authors provide the number of replicates used for the barcharts - best within the chart.

All data is shown (4 replica for trypsin).

Reviewer #3:

An integrated workflow for crosslinking mass spectrometry

Marta L. Mendes#, Lutz Fischer#, Zhuo A. Chen, Marta Barbon, Francis J. O'Reilly, Sven Giese, Michael Bohlke-Schneider, Adam Belsom, Therese Dau, Colin W. Combe, Martin Graham, Markus R. Eisele, Wolfgang Baumeister, Christian Speck, Juri Rappsilber

The authors describe a method for the consecutive digestion of cross-linked protein-complexes using different proteases and report an updated version of their database search engine 'Xi'. The authors use the properties of different proteases in combination and compare tryptic peptides to peptides derived from parallel digestion and sequential digestion of trypsin with either AspN, GluC or chymotrypsin. The number of unique identifications increases by combining peptides obtained from trypsin digest with peptides from the sequential digest with trypsin and either AspN, GluC or chymotrypsin. The bioinformatic analysis of the mass spectrometric data from parallel and sequential digestion is achieved by a newly developed feature of their search algorithm Xi. I appreciate the demonstrated benefit of sequential digestion for crosslinking analysis, however, the technical advantage of consecutive digests has been published before as well as has been the Xi software for non-functionalized crosslinkers. Neither the technical developments nor the interpretation of the crosslink information on structural models provide sufficient novelty to justify publication of this manuscript in *Molecular Systems Biology*. Besides this, I am wondering whether a report on a technical development in the crosslinking field fits the scope of *Molecular Systems Biology*. As this work is a valuable contribution to the development of crosslinking workflows, I recommend transferring this manuscript to a more specialized analytical/proteomics journal.

While sequential digestion forms an important part of the novelty of our work and is discussed below, we see a number of additional results presented by our work:

- Sequential digestion outperforms parallel digestion and trypsin replicates as it generates smaller tryptic peptides, without overshooting them, facilitating the identification of crosslinked peptides.
- The workflow has a broad applicability: it can be used to study protein dynamics, topology of protein complexes, conformational changes (qCLMS) and uses homo and heterobifunctional crosslinkers.
- We were able to detect a dynamic interaction in the OCCM complex not seen before by EM which can be a potential target for cancer therapy.
- XiSearch supports searches with specificity to preferentially Lysine followed by Serine, Threonine and Tyrosine, when using BS3.

Note that this manuscript is our lab's presentation of sequential digestion and xiSEARCH. One paper by our group uses sequential digestion but references to this manuscript in BioRxiv. The search strategy of xiSEARCH has been described but not the tool (and its open source code).

General Comments:

1. The finding of the Authors was anticipated since various research articles including their own work describe similar approaches:

- a. Dau T, Gupta K, Berger I, Rappsilber J. Sequential digestion with Trypsin and Elastase in cross-linking/mass spectrometry. *Analytical Chemistry* 2019 91 (7): 4472-4478
- b. Wiśniewski J and Mann M. Consecutive Proteolytic Digestion in an Enzyme Reactor Increases Depth of Proteomic and Phosphoproteomic Analysis. *Analytical Chemistry* 2012 84 (6): 2631-2637
- c. Gilmore JM, Kettenbach AN, Gerber SA. Increasing phosphoproteomic coverage through sequential digestion by complementary proteases. *Anal Bioanal Chem.* 2011;402(2):711-720.
- d. Guo X, Trudgian DC, Lemoff A, Yadavalli S, Mirzaei H. Confetti: a multiprotease map of the HeLa proteome for comprehensive proteomics. *Mol Cell Proteomics.* 2014;13(6):1573-1584.
- e. Larsen MRI, Højrup P, Roepstorff P. Characterization of gel-separated glycoproteins using two-step proteolytic digestion combined with sequential microcolumns and mass spectrometry. *Mol Cell Proteomics.* 2005 Feb;4(2):107-19.
- f. And others

We have added a section in Discussion on the novelty of sequential digestion:

“Sequential digestion novelty

The use of proteases other than trypsin to achieve complementary information in protein analysis dates back to at least 1987 (Aebersold *et al*, 1987). Since then, several works used parallel digestion and fewer used sequential digestion to increase sequence coverage and/or complementarity in simple protein complexes (Mohammed *et al*, 2008), proteomes (MacCoss *et al*, 2002; Swaney *et al*, 2010; Guo *et al*, 2014), phosphoproteomes (Giansanti *et al*, 2015; Gilmore *et al*, 2012; Wang *et al*, 2008) and other post-translational modifications (Larsen *et al*, 2005). In CLMS, parallel digestion was reported first by Pinkse *et al*. in 2009 to increase complementarity and target a specific cross-link site (Pinkse *et al*, 2009). Leitner *et al*. in 2012 presented cross-link data obtained when using in parallel five different proteases on a mix of standard proteins. Unfortunately, it remained unclear if the parallel use of five proteases would have been outperformed by simply five-times re-analysing a tryptic digest. Re-analysing tryptic digests results in additional crosslinks being detected (Müller *et al*, 2018) also seen here (analysing a trypsin digest four-times increased the observation from 167 to 251 unique residue pairs (URPs), Fig. 1B,C). We then analytically prove that parallel digestion outperforms repeated injection of tryptic digests for cross-link detection, albeit moderately (261 versus 251 URPs, Fig. 1C). Importantly, we expand this by showing that sequential digestion is even better (339 URPs, Fig. 1C).

We underpin these observations by proposing a mechanistic model that explains why sequential digestion outperforms parallel and repeated use of proteases. Outperforming repeated use of trypsin is linked to sequential-digested crosslinked peptides being smaller (Fig. EV2A) which in fact improves detection rates as most observed peptides fall between 1000-2000 Da (Fig EV3D). The second digestion targets primarily long peptides as can be seen both in the surviving long peptides (poor in secondary cleavage sites) as well as in the observed short peptides (rich in secondary cleavage sites) (Fig. EV3). Consequently, sequential digestion is effective and does not shorten crosslinked peptides such that they become less observable (below 1000 Da). Outperforming parallel digestion is linked to the detection bias of proteomics towards peptides with tryptic C-termini, that has been noted for linear peptides (Giansanti *et al*, 2016). Cleaving a crosslinked peptide N-terminal of the crosslink site maintains the tryptic C-termini while cleaving C-terminal leads to a non-tryptic C-terminus. During digestion, both should be equally likely. However, the former will be more likely to be detected which is reflected in our data (Fig. EV1C). As all crosslinked peptides of parallel digestion other than those of trypsin use are lacking tryptic C-termini parallel digestion has a marked disadvantage. “

Accepted

21st August 2019

Thank you for sending us your revised manuscript. We are now satisfied with the modifications made and I am pleased to inform you that your paper has been accepted for publication.

USEFUL LINKS FOR COMPLETING THIS FORM

Corresponding Author Name: Juri Rappsilber
Journal Submitted to: Molecular Systems Biology
Manuscript Number: